# Sustainability on Different Canola (*Brassica napus* L.) Cultivars by GGE Biplot Graphical Technique in Multi-Environment

Seyed Habib Shojaei [1], Khodadad Mostafavi [2], Seyed Hamed Ghasemi [1], Mohammad Reza Bihamta [3], Árpád Illés [4], Csaba Bojtor [4,*], János Nagy [4], Endre Harsányi [4], Attila Vad [5], Adrienn Széles [4] and Seyed Mohammad Nasir Mousavi [4,*]

1   Department of Biotechnology and Plant Breeding, Science and Research Branch, Islamic Azad University, Tehran 14778-93855, Iran; habibshojaei@nigeb.ac.ir (S.H.S.); hamed.ghasemi@srbiau.ac.ir (S.H.G.)
2   Department of Agronomy and Plant Breeding, Karaj Branch, Islamic Azad University, Karaj 31499-68111, Iran; mostafavi@kiau.ac.ir
3   College of Agriculture & Natural Resources (UCAN), University of Tehran, Karaj 31499-68111, Iran; mrghanad@ut.ac.ir
4   Institute of Land Use, Engineering and Precision Farming Technology, Faculty of Agricultural and Food Sciences and Environmental Management, University of Debrecen, 138 Böszörményi St., 4032 Debrecen, Hungary; illes.arpad@agr.unideb.hu (Á.I.); nagyjanos@agr.unideb.hu (J.N.); harsanyie@agr.unideb.hu (E.H.); szelesa@agr.unideb.hu (A.S.)
5   Institutes for Agricultural Research and Educational Farm (IAREF), Farm and Regional Research Institutes of Debrecen (RID), Experimental Station of Látókép, 4032 Debrecen, Hungary; vadattila@agr.unideb.hu
*   Correspondence: bojtor.csaba@agr.unideb.hu (C.B.); nasir@agr.unideb.hu (S.M.N.M.)

**Abstract:** Knowledge about the extent of genotype in environment interaction is helpful for farmers and plant breeders. This is because it helps them choose the proper strategies for agricultural management and breeding new cultivars. The main contribution of this paper is to investigate genotype on environmental interaction using the GGE biplot method (Genotype and the Genotype-by-Environment) in ten canola cultivars. The experimental design was a randomized complete block design (RCBD) with three replications to assess the stability of grain yield of ten canola cultivars in five regions of Iran, including Birjand, Karaj, Kashmar, Sanandaj, and Shiraz, within two agricultural years of 2016 and 2017. The results of combined ANOVA illustrated that the effects of the environment, genotype × environment, and genotype were highly significant at 1%. Variance Analysis showed that three environmental impacts, genotype, and interaction of genotype in the environment effect, produced 68.44%, 18.63%, and 12.9% of the total variance. The GGE biplot graphs were constructed using PCA. The first principle component (PC1) explained 65.3%, and the second (PC2) explained 18.8% of the total variation. The research examined polygon diagrams to identify two top genotypes and four mega-environments. Also, the appropriate genotypes for each environment were diagnosed. Using the GGE biplot, it was possible to make visual comparisons and identify superior genotypes in canola. Accordingly,. The results obtained from graphical analysis indicated that Licord, Hyola 401 and Okapi genotypes showed the highest yield and were selected as the most stable genotypes. Also, Karaj region was chosen as a experimental region where the screening of genotypes was very suitable. Based on the ranking of the genotypes in the most suitable region (Karaj), Okapi genotype was selected as the desired genotype. In examining the heatmap drawn between the genotypes and the investigated environments, a lot of similarity between the genotypes of Sarigal, Hyola 401 and Okapi was observed in the investigated environments. The GGE biplot graphs enabled the detection of stable and superior environments and the grouping of cultivars and environments based on grain yield. The results of this research can be used both for extension and for future breeding programs. Our results provide helpful information about the canola genotypes and environments for future breeding programs.

**Keywords:** canola; combined analysis; genotype in environment interaction; GGE biplot

## 1. Introduction

Plants in the Brassicaceae family, such as rapeseed, are cultivated under various climatic conditions. There are numerous uses for rapeseed, including feed, grazing, forage, and industrial applications. As a result of the characteristics of this plant, it has been extensively cultivated in many parts of the world due to its many benefits. However, a suitable cultivar should be cultivated in each area to achieve the best performance. In this regard, grain yield stability is considered one of the selection criteria for selecting the best cultivar for each condition and climate [1]. Despite the diverse climatic conditions in Iran and rapeseed cultivation in different locations, more research needs to be done on the yield stability of rapeseed cultivars. High-yielding and stable cultivars are necessary as Iran imports vast amounts of oilseeds annually. Stable varieties are known to exhibit high yield and persistent performance in a range of environments. Highly adapted cultivars in various environments usually have moderate and stable yields. However, cultivars that only have good genetic potential in favorable conditions and perform poorly in adverse conditions are considered limited adaptation cultivars [2,3]. The seed yield of canola is a quantitative trait primarily influenced by the different environmental effects; hence, in most cases, it has low heritability [4].

GGE biplot (Genotype and the Genotype-by-Environment) visualizes varieties' performance rank and stability across environments for decision-making regarding releasing new cultivars [5]. Measuring the amount of genotype in environment interaction helps breeders to evaluate genotypes more accurately and to select superior genotypes with stability and high yield [6]. Genotype in environment interaction make it difficult to determine the exact contribution of new cultivars, favorable environment, and agricultural machinery to yield [7]. Numerous statistical techniques have been presented to study genotype in environment interactions and to define stable genotypes. These methods are divided into two main univariate and multivariate groups [8]. Among the multivariate methods, the GGE biplot is a favorite method for researchers based on principal component analysis [9,10]. The GGE biplot method provides a premium solution for combining average performance and sustainability into a chart that can be evaluated graphically and visually to comprehend Genotype × Environment (GE) interaction [11,12]. The GGE biplot is usually highly practical in breeding programs for each crop. The environments are grouped into several groups using the GGE biplot. The environmental response in each group to the evaluated genotypes is approximately similar [13,14].

Yan et al. [15,16] showed that the performance and accuracy of the GGE biplot model were higher than that of the additive main-effects and multiplicative interaction (AMMI) model. Rahnejat et al. [17] identified Okapi, Modena, and GKH 305 genotypes as high-yield and adaptable genotypes on 15 rapeseed genotypes in four different regions of Iran. Alizadeh et al. [18] investigated the genotype × environment effect to select winter rapeseed varieties with a high yield on 11 rapeseed genotypes. They showed that the most favorable genotypes were selected and reported regarding seed yield traits. In another study by Mousavi et al. [19], GGE biplot and AMMI methods were investigated regarding seed yield traits and oil percentage in sunflower cultivars. The results of this study showed that Hyola 401, Okapi, and Sarigol cultivars had high yields in terms of seed yield and Option 500 and Sunday cultivars in terms of oil percentage.

The current study analyzed genotype × environment interactions for 15 canola cultivars using GGE Biplot. This study desired to determine the stable genotypes on grain yield and suggest the most suitable genotype (s) for different environments.

## 2. Materials and Methods

### 2.1. Experiment Design

This study evaluated the yield stability of ten canola cultivars (Table 1). Five regions were tested by randomized complete block design (RCBD) with three replications. In 2016 and 2017, these regions included Birjand, Karaj, Kashmar, Sanandaj, and Shiraz. As shown in Table 2, the study areas are geographically characterized by the following characteristics.

In all areas, agricultural operations were carried out in the same manner. Each experimental plot contained four rows separated by 50 cm, each of which was of a length of 4 m. Regular irrigation, weeding, and thinning were carried out at all stages of plant growth. Before flowering, irrigation is performed every seven days, and after flowering, irrigation is performed every 12 days. After physiological maturity, the grain yield of each genotype was measured in all experiments. For this purpose, two rows in the middle were used to remove the one-half meter from the beginning and end of the lines. The ripening of the seeds starts from the lower part of the stem and spreads upwards, and it is not wise to wait until the plants are completely ripe and dry before harvesting.

**Table 1.** Canola cultivar names and codes.

| Genotype No. | Genotype | Origin | Genotype No. | Genotype | Origin |
|---|---|---|---|---|---|
| G1 | Sarigol | Iran | G6 | Likord | Germany |
| G2 | Hyola308 | Canada | G7 | Okapi | France |
| G3 | Option500 | Germany | G8 | Hyola401 | Canada |
| G4 | Opera | Sweden | G9 | Zarfam | Iran |
| G5 | Modena | Denmark | G10 | Modena | Denmark |

**Table 2.** Geographical specifications of areas where the experiments.

| Area | Longitude | Latitude | Elevation AMSL (m) | Temperature (°C) | Rainfall Average (2016–2017) | EC(ds/m) | Acidity | Lime (%) | Organic Carbon (%) | Organic Materials (%) | Clay (%) | Silt (%) | Sand (%) |
|---|---|---|---|---|---|---|---|---|---|---|---|---|---|
| Karaj | 50°54′ E | 35°55′ N | 1312 | 18.3 | 288.5 | 0.20 | 8.2 | 7 | 32 | 45 | 32 | 25 | 22 |
| Birjand | 59°12′ E | 32°52′ N | 1491 | 21 | 143.95 | 0.5 | 7.4 | 8 | 25 | 35 | 42 | 18 | 31 |
| Shiraz | 52°36′ E | 29°32′ N | 1484 | 17 | 328.9 | 0.26 | 7.22 | 6 | 42 | 53 | 34 | 28 | 16 |
| Kashmar | 58°48′ E | 35°53′ N | 1109 | 19 | 198 | 0.32 | 7.88 | 7 | 36 | 51 | 31 | 24 | 24 |
| Sanandaj | 47°00′ E | 35°20′ N | 1373 | 16 | 461 | 0.27 | 7.45 | 7 | 40 | 46 | 36 | 22 | 24 |

## 2.2. Analysis Method

A first test was conducted on the data collected from each site to determine if there was homogeneity of error variances between experiments. Then we conducted a combined analysis of variance (ANOVA) following data collection from each site. To visualize the multivariate graphical GGE biplots, a singular value decomposition method was applied following the equation below.

$$Y_{ij} - \mu - \beta_j = \lambda_1\ \xi_{i1}\ \eta_{j1} + \lambda_2\ \xi_{i2}\ \eta_{j2} + \varepsilon_{ij} \tag{1}$$

where, $Y_{ij}$ is the mean of ith genotype in jth environment, $\mu$ is the mean of genotypes, $\beta_j$ is the mean influence of the jth environment, $\lambda_1$ and $\lambda_2$ are the special values for the first and second components, $\xi_{i1}$ and $\xi_{i2}$ are specific genotypic vectors, and $\eta_{j1}$ and $\eta_{j2}$ are the environmental vectors of the first and second components, and $\varepsilon_{ij}$ is the residual value for the ith genotype in the jth environment [20].

Analysis of variance and the Bartlett test were performed with SAS version 9.1. To perform a combined variance analysis, the environment and genotype were treated as random factors and the genotype as a constant factor. The GGE biplot Version 0.1.3 [21] software is based on six patterns:

1. Evaluating together the grain yield and the stability of two cultivars at the same time
2. Determine the most suitable genotype in each environment
3. Evaluating the relationships between genotypes
4. Ranking of genotypes based on the most suitable environment
5. Classification of environments based on the most suitable genotype
6. Appraising of relationships between environments using graphical analysis of GGE biplot

## 3. Results and Discussion

### 3.1. Variance Analysis

The homogeneity of variance error test confirmed the uniformity of experimental error variances. Combined variance analysis illustrated that the effect of environment, genotype, and the interaction of genotype × environment was highly significant ($p < 0.01$). Consequently, it is possible to group settings based on interaction. The diversity in the response of canola cultivars to different environments shows that it is possible to apply genetic diversity to improve grain yield [22]. Grain yield is a quantitative trait controlled by several factors such as plant density, number of pods per plant, number of grains per pod, and grain weight. Therefore, grain yield has high variability and depends on cultivar, environment, and interaction [22].

As shown in Table 3, 68.44% of the total variation caused by location, genotype, and GL belonged to location. The proportion of genotype and genotype × environment interaction was 18.6% and 12.9%, respectively. In a trial evaluating six canola cultivars in different parts of India, it was shown that the proportion of variance caused by the environment, genotype, and the interaction of genotype × environment was 78.7, 7.6, and 13.6%, respectively [22]. However, ANOVA revealed that out of the total variation made from genotype, year, and GY interaction, the proportion of genotype variance was much higher than other factors. According to Table 3, the GY interaction was more significant than the genotype, suggesting that there are multiple mega-environments in the canola-growing areas of Iran. These mega-environments are portions of plant species grown in homogeneous conditions, resulting in similar yield performance for some genotypes [23].

**Table 3.** Five environments and two years of combined analysis of variance for canola cultivars.

| Source of Variation | df | Sum of Squares | Mean Square | % of L + G + GL | % of Y + G + GY | *p* Value |
|---|---|---|---|---|---|---|
| Location (L) | 4 | 212.50 | 53.12 ** | 68.44 | | $p < 0.001$ |
| Year (Y) | 1 | 0.78 | 0.78 ** | | 8.69 | $p < 0.001$ |
| Location × Year (L × Y) | 4 | 277.55 | 0.69 ** | | | $p < 0.001$ |
| Rep/(Loc × Year) | 20 | 11.55 | 0.57 | | | |
| Genotype (G) | 9 | 57.87 | 6.43 ** | 18.63 | 58.60 | $p < 0.001$ |
| Location × Genotype (L × G) | 36 | 40.08 | 1.11 ** | 12.91 | | $p < 0.001$ |
| Year × Genotype (Y × G) | 9 | 32.28 | 3.58 ** | | 32.69 | $p < 0.001$ |
| Location × Year × Genotype | 36 | 137.74 | 3.82 ** | | | $p < 0.001$ |
| Error | 299 | 140.39 | 53.12 | | | |

**: Significant at 1%.

### 3.2. AEC View

Due to the significant genotype in environment interaction, combined variance analysis cannot describe the stability of genotypes. Therefore, GGE biplot graphical analysis was used based on the principal component analysis (PCA) method to determine the stable genotypes and investigate the between genotype and environment interaction. The PCA method increases our awareness of the interaction between genotype and environment and overcomes the limitations of the univariate ANOVA method, and allows the breeders to select the best genotypes [24]. This method reveals the response of each cultivar in different environments [25]. The results of the GGE biplot method showed that the first and second principal components explained 65.1% and 18.8% changes in grain yield, respectively (in total 84.1%). The sum of the first two components explains at least 60% of the data diversity; this model has satisfactory validity [26].

Using the average-environment coordination (AEC) view of the GGE biplot, we were able to simultaneously compare the yields of the cultivars in terms of yield and stability (Figure 1). In this figure, the axis with a specified arrow (AEC-abscissa) and the resulting mean values (circles) determine the performance of the digits, so that each digit to the right of this axis has more performance. The axis marked with two arrows (AEC-ordinate) confirms the stability or instability of canola cultivars. Genotypes distant from this axis's

origin (AEC-abscissa) have less stability [27,28]. The G8 and G7 cultivars are described as having higher grain yields than other cultivars and being stable, making them desirable genotypes.. Genotype G6 had the least distance from the horizontal axis but was weak in grain yield and was not classified as the preferred genotype.

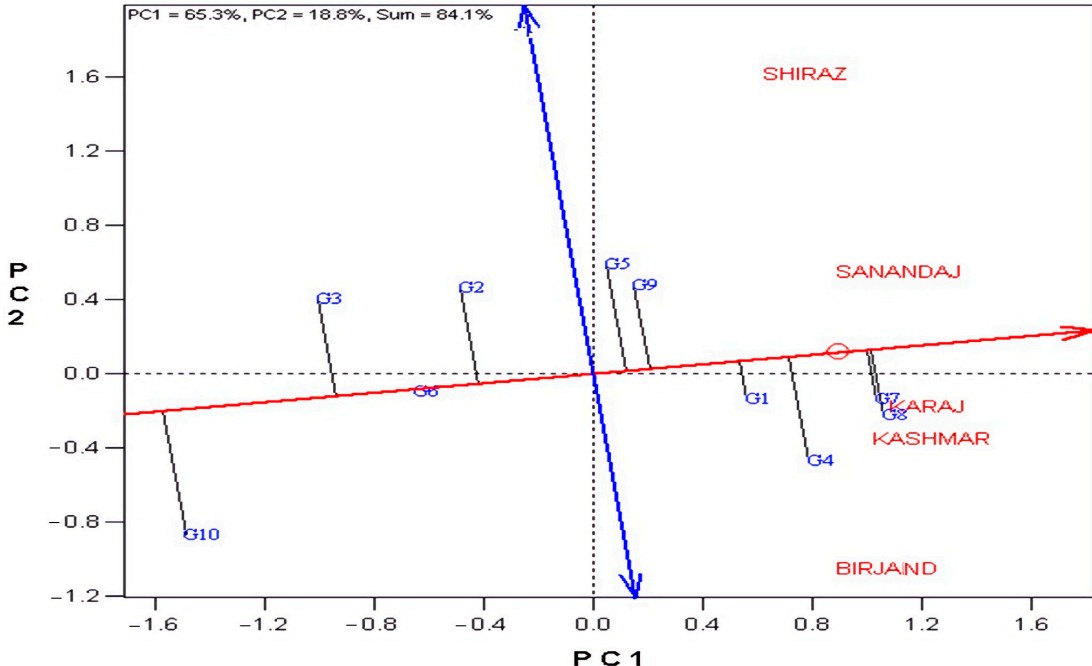

**Figure 1.** Evaluation of 10 canola cultivars across five environments using the average environment coordination (AEC) view, which classifies cultivars simultaneously based on grain yield and yield stability. (G1: Sarigol, G2: Hyola308, G3: Option500, G4: Opera, G5: Sunday, G6: Likord, G7: Okapi, G8: Hyola401, G9: Zarfam, G10: Modena).

Furthermore, G10 (Modena) was identified as the most unstable genotype with the most significant distance from this axis (Figure 1). The order of genotypes from the most desirable genotype to the most unfavorable genotype is G8-G7-G4-G1-G9-G5-G2-G6-G3-G10. It is important to note that this method needs to differentiate the genotypes completely. Since this method has no hypothesis test, the conclusions are based on visual inspection [29].

*3.3. Polygon View*

The GGE biplot was plotted To investigate the best genotypes in each environment [30]. This kind of data biplot allows us to identify the best genotypes in each environment and detect stable genotypes in all environments [31]. According to Figure 2, the graph is divided into six sections. This graph shows that the genotypes G4, G7, G8, G9, G5, G3, and G10 are located farthest from the biplot origin. The cultivars at the polygon's corner can be called vertex cultivars. These cultivars are the most responsive to the environment. Shiraz is located in the segment where the G5 and G5 genotypes are located, which means that these genotypes have the best performance in Shiraz. Karaj and Kashmar zones are found in the G8 and G7 genotypes, so G8 and G7 genotypes were the best for Karaj and Kashmar. After these two genotypes, the G1 genotype was the best compared to other regions' genotypes. Sanandaj is located where there are no genotypes in this sector, which indicates that the existing genotypes have yet to be able to perform well in this area. G4 is recognized as the best genotype in Birjand. Also, the G6 genotype, located in the center of the plot, showed a similar response to the environments studied, meaning that this genotype is generally adapted to all environments. The environments studied here were divided into mega-environments. The first mega-environment includes Karaj and Kashmar in central

Iran. After that, Shiraz is the best region in the south of Iran; then Sanandaj is a suitable region in the west of Iran, and Birjand is the best region in the east of Iran. However, it has been observed that due to the high interaction between year and location, for some crops, such as wheat, the which-won-where diagram is not repeatable; hence, it is impossible to separate the planting environments as a real mega-environment [29]. The unique feature of the GGE biplot is that graphs can show which environment or subgroup is the best cultivation area for each cultivar [14].

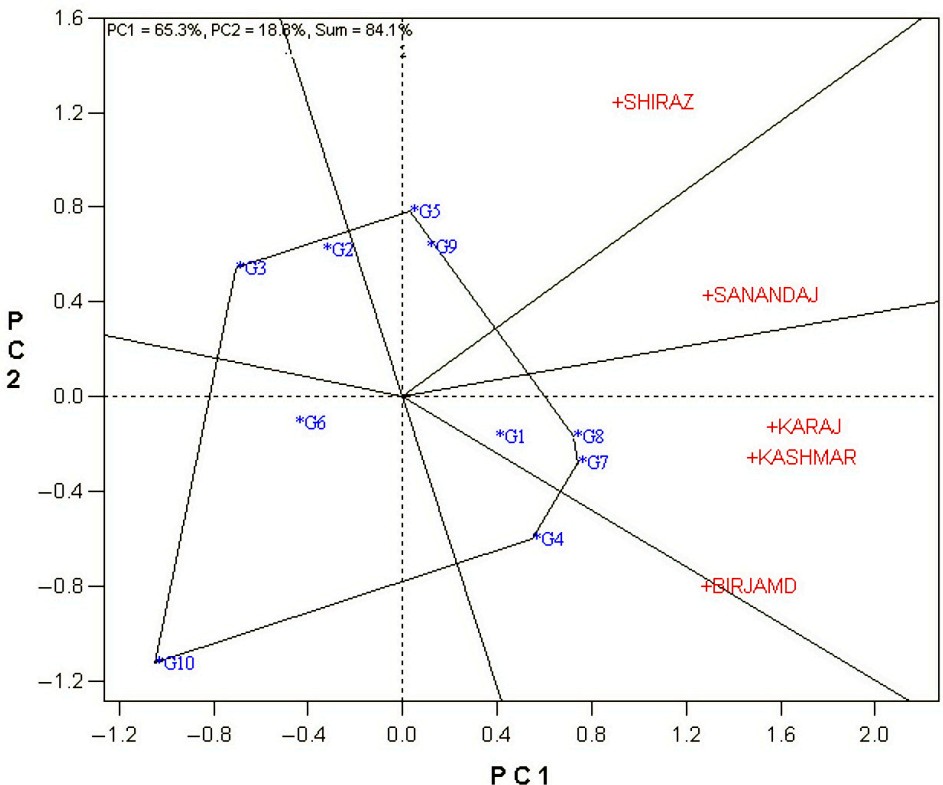

**Figure 2.** Polygon views of the GGE biplot based on symmetrical scaling for the genotypes and environments. (*G1: Sarigol, *G2: Hyola308, *G3: Option500, *G4: Opera, *G5: Sunday, *G6: Likord, *G7: Okapi, *G8: Hyola401, *G9: Zarfam, *G10: Modena).

*3.4. Genotype Grouping*

Figure 3 shows the correlation and relationships between genotypes. In this diagram, as the angle between the vectors of genotypes is smaller, there is more correlation between them. Conversely, if the angle of the vectors is greater than 90 degrees, it shows a negative correlation between genotypes [32,33]. Accordingly, the genotype is divided into four groups. The first group consisted of G8, G7, G1, and G4 genotypes, the second group included G9 and G5, and the third group could be assigned to G2 and G3 genotypes. The G6 and G10 genotypes were classified into the fourth group. There was a high correlation between the genotypes within each group, indicating similar responses to different environments, so there was no significant difference between the performance ratings of these genotypes in different environments. G10, for instance, has an angle greater than 90 degrees with G5, G9, G8, G7, G1, and G4 genotypes and, as a result, has a negative correlation with these genotypes. Therefore, in breeding programs based on hybridization, it is recommended to use parental genotypes from different groups to do hybridization and develop breeding populations. In one study, different rapeseed cultivars were evaluated in five regions of Chile [16]. This study stated that the genotype × environment interaction significantly explained total variance. The first and second principal component analyses (PC1, PC2) accounted for 74.5% of the total variance [16].

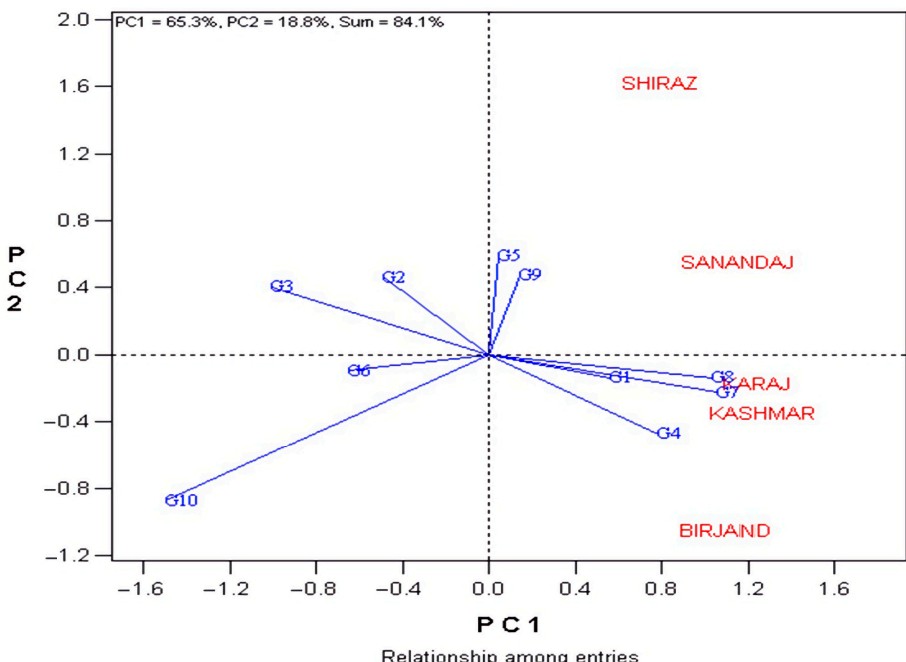

**Figure 3.** Correlation between canola cultivars for grain yield measured at 5 locations using biplot diagrams. (G1: Sarigol, G2: Hyola308, G3: Option500, G4: Opera, G5: Sunday, G6: Likord, G7: Okapi, G8: Hyola401, G9: Zarfam, G10: Modena).

### 3.5. Ranking of Genotypes Based on the Most Suitable Environment

Figure 4 indicates the environment's ranking based on the best genotype. On this biplot, the HYOLA401 genotype is the best genotype to introduce in the Karaj environment in this research. Shiraz was the most unfavorable for its cultivation. The ranking of environments based on this cultivar is as follows: Karaj > Birjand > Kashmar > Sanandaj > Shiraz.

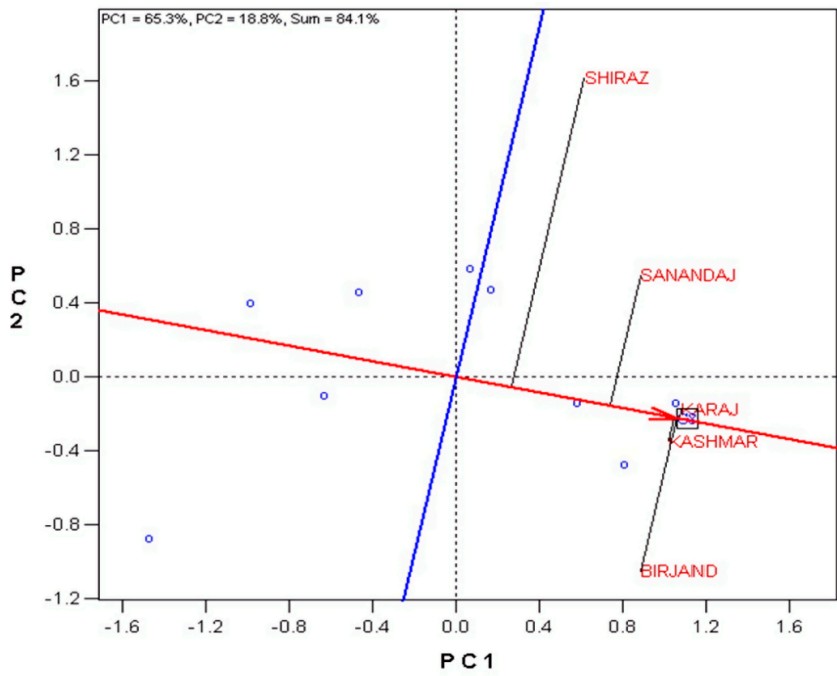

**Figure 4.** Ranking of environments based on cultivars in GGE biplot graph, G8 (Hyola401). (G1: Sarigol, G2: Hyola308, G3: Option500, G4: Opera, G5: Sunday, G6: Likord, G7: Okapi, G8: Hyola401, G9: Zarfam, G10: Modena).

### 3.6. Ranking of Genotypes Based on the Most Suitable Environment (Karaj)

Several breeding companies, both private and public, are located in Karaj. Figure 5 shows the ranking of genotypes according to grain yield in Karaj. This way, the linear coordinates are connected to the desired location, Karaj, extending to both sides, called the peripheral axis [33]. The vertical lines represent the performance ranking of 10 genotypes in the center of Karaj. The genotypes oriented toward the positive end-points of this axis have satisfactory performance in this region. Still, genotypes oriented toward the negative end in the graph have poor performance in Karaj. Based on this chart, the ranking of genotypes is as follows: G10 < G3 < G6 < G2 < G5 < G9 < G1 < G4 < G7 < G8.

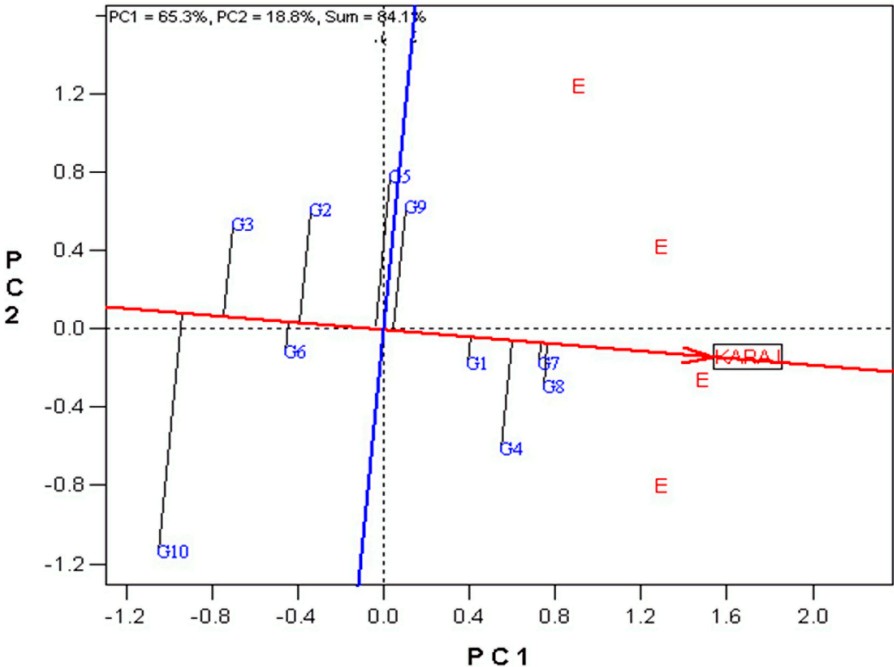

**Figure 5.** Biplot graph of GGE canola cultivars ranked according to grain yield in the best environment (G1: Sarigol, G2: Hyola308, G3: Option500, G4: Opera, G5: Sunday, G6: Likord, G7: Okapi, G8: Hyola401, G9: Zarfam, G10: Modena).

### 3.7. Relationships between Environments

Correlations between environments can reveal relationships between environments. It is also essential for future experiments. An experiment cannot be carried out in a correlated environment if two or more environments correlate [1,34]. The Cosine of the angles between the vectors of the environments represents correlation, and vector length indicates how powerful the environments are in discriminating genotypes. If the angle between the axis of the two environments is acute, the correlation between the two environments will be positive. If the angle between the axis of the two environments is obtuse, the correlation between those two environments will be found to be negative [35]. Positive correlations between Shiraz-Sanandaj, Shiraz-Karaj, Shiraz-Kashmar, Sanandaj-Karaj, Sanandaj-Kashmar, Sanandaj-Birjand, Karaj-Kashmar and Kashmar-Birjand. This suggests a similar response to genotypes in these locations. Therefore, there will be no significant difference between the performance ranks of cultivars in these locations. However, considering that the angle between Shiraz and Birjand vectors is close to 90 degrees, it means that the correlation between these locations is almost zero, which is indicative of the difference between these areas in the production of rapeseed (Figure 6). In other words, genotypes in these two locations have independent function trends. Changes between environments can be due to climate change, soil, and other changes in environments and other fluctuations during years [36,37].

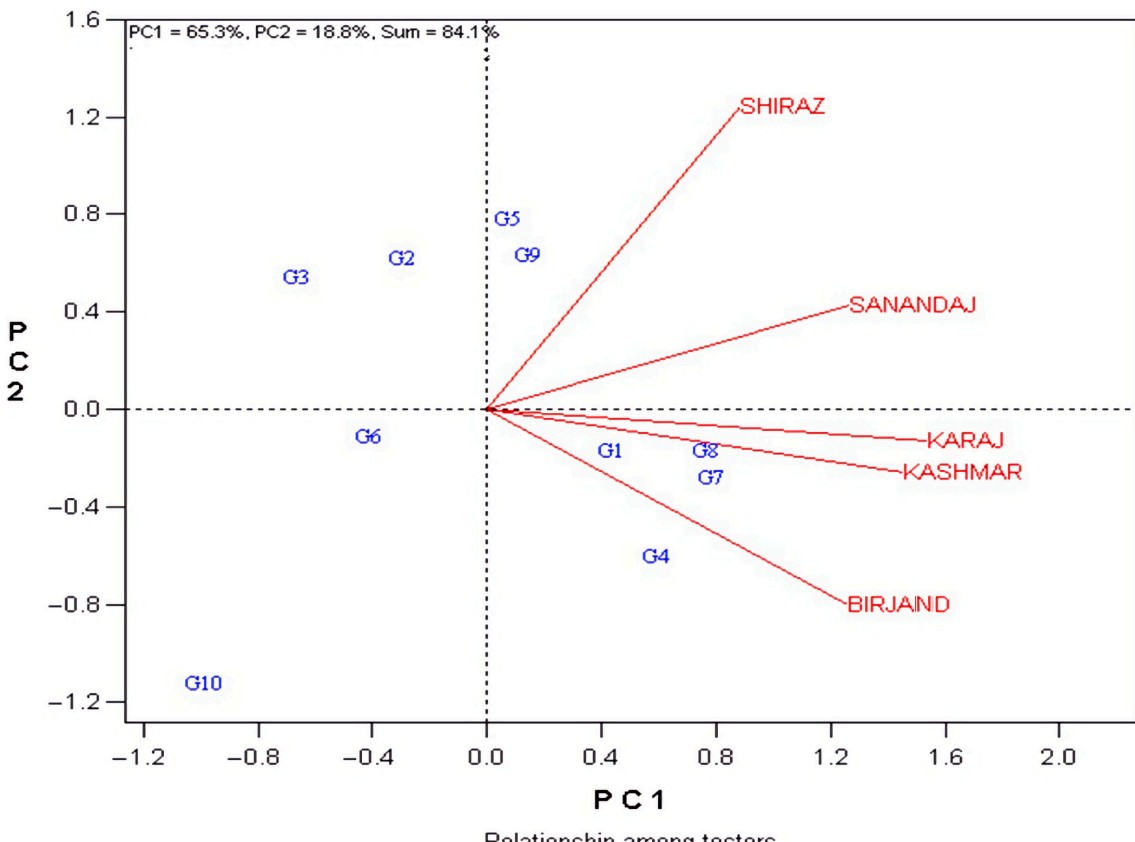

**Figure 6.** Correlation between environment and grain yield in five trials of canola cultivars. (G1: Sarigol, G2: Hyola308, G3: Option500, G4: Opera, G5: Sunday, G6: Likord, G7: Okapi, G8: Hyola401, G9: Zarfam, G10: Modena).

Another essential feature of correlation between environments is the environment vector length, which is an approximation of standard deviation within each environment and an indicator of the differentiation ability. The differentiation capability in an environment has been an important feature, so environments lacking the ability to differentiate cannot provide helpful information about the cultivars [38].

### 3.8. Ideal Environment with the Ranking Biplot

The environment rating was based on the hypothetical ideal environment for the average data of two years of the experiment (Figure 7). According to this diagram, any environment near the hypothetical superior environment is more desirable than other environments. In this graph, the average environment is shown with a small circle, which is determined by using the average of the first and second components. The horizontal axis shows the performance of environment and the vertical axis is an estimate of the interaction of genotype × environment of each environment. The environment at the furthest distance from the hypothetical perfect environment will be undesirable. As a result, the environment ranking is as follows:

Karaj > Sanandaj = Kashmar > Birjand > Shiraz.

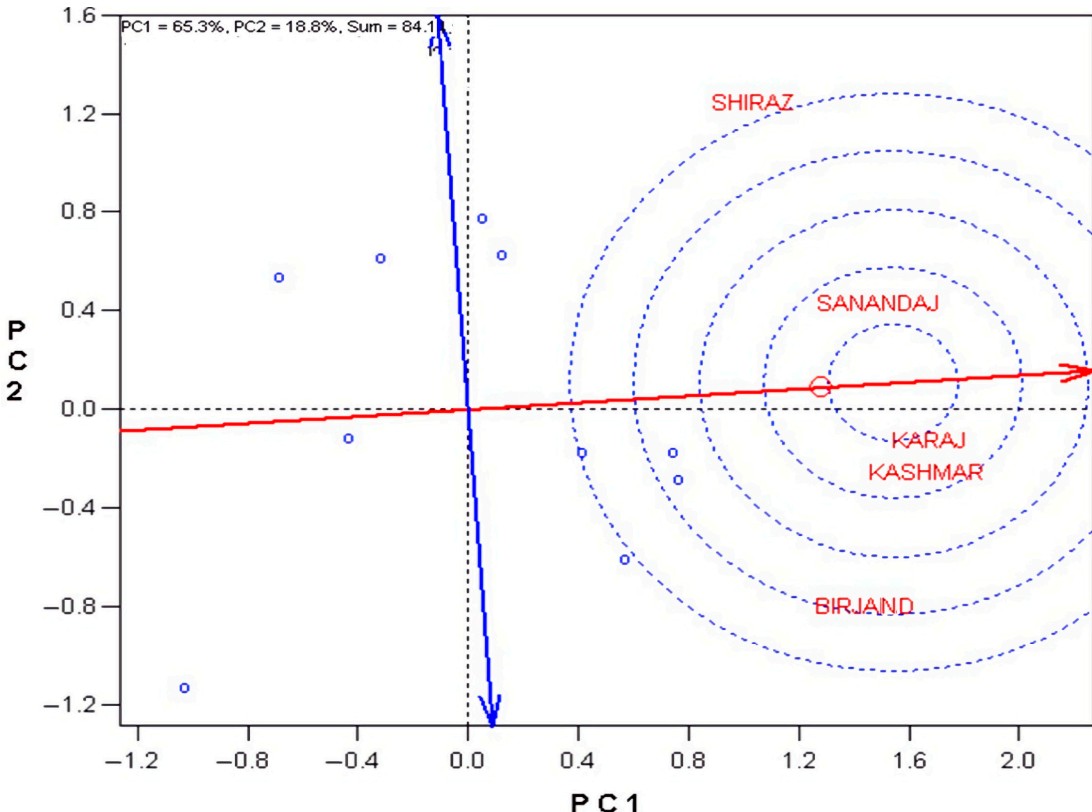

Ranking testers based on both discriminating ability and representativeness

**Figure 7.** Comparing the ideal environment with the ranking biplot. (G1: Sarigol, G2: Hyola308, G3: Option500, G4: Opera, G5: Sunday, G6: Likord, G7: Okapi, G8: Hyola401, G9: Zarfam, G10: Modena).

*3.9. Ideal Genotype Based on Grain Yield and Stability Simultaneously*

Identifying the hypothetical ideal genotype with high stability and yield is possible. The ideal and desirable genotype will have the most yield and highest stability. Any genotype most relative to this model is the most desirable one. The one with the most distance is introduced as the most undesirable genotype [38]. All other genotypes can be ranked according to their distance from the ideal genotype [38]. The AEC ordinate is the double-arrowed line that passes through the biplot origin and is perpendicular to the AEC abscissa (Figure 8). In this graph, the average genotype is shown with a small circle, which is determined by using the average of the first and second components. The horizontal axis shows the performance of genotypes and the vertical axis is an estimate of the interaction of genotype × environment of each. Therefore, G10 near the bottom of the biplot is more variable and less stable than other cultivars. The small circle located on the AEC abscissa with an arrow pointing to it represents the ideal cultivar in Figure 8.

Two criteria define it:

(1) It has the highest yield of the entire dataset.
(2) It is stable, as indicated by being located on the AEC abscissa.

Such an ideal genotype.
G8 > G7 > G1 > G4 > G9 > G5 > G2 > G6 > G3 > G10.

Choosing the suitable selection method requires minimizing the interaction between genotype and environment, especially for farmers and plant breeders. This study was conducted to understand the genotype × environment interaction in canola using a GGE biplot. The result of the GGE biplot method interpreted for stability analysis of cultivars regarding grain yield. This method demonstrated the differences and similarities between cultivars and environments, as well as the interaction between the genotype and the environment, based on a similar method reported by [33].

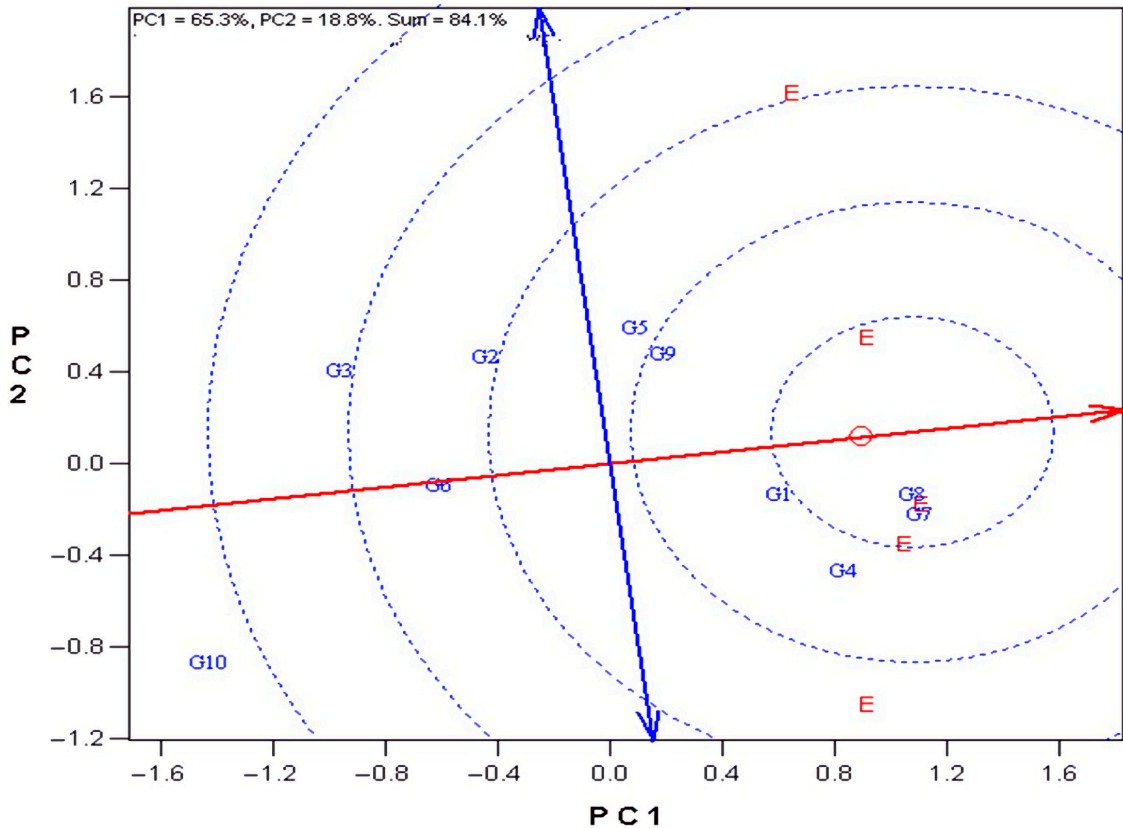

**Figure 8.** GGE-biplot for comparison of the canola cultivars with the ideal genotype based on grain yield and stability simultaneously. (G1: Sarigol, G2: Hyola308, G3: Option500, G4: Opera, G5: Sunday, G6: Likord, G7: Okapi, G8: Hyola401, G9: Zarfam, G10: Modena).

Nevertheless, it may be helpful as a reference for cultivar evaluation. A cultivar's desirability is measured based on the plot distance between it and this ideal cultivar. The concentric circles are used to visualize the distance between all cultivars and the ideal cultivar (Figure 8). Hence, as shown in Figure 8, G8 and, subsequently, G7 were identified as the best genotypes given that they have the least distance to the ideal hypothetical genotype and genotype G10 was recognized as the most undesirable genotype as it has the most significant distance from the ideal hypothetical genotype.

*3.10. Cluster Analysis by Heat Map Method*

Cluster analysis based on the average effect of genotype × environment in the two years of the experiment clustered the genotypes into three main groups and the studied environments into three main groups. Based on the clustering of genotypes, G3, G5, G2, and G10 genotypes were placed in the first group, G8, G1, and G7 genotypes in the second group, and G4, G6, and G9 genotypes in the third group and selected as desirable genotypes regarding the investigated environments on grain yield traits. Also, in the grouping of environments, Sanandaj and Karaj were placed in the first group, the Kashmar environment in the second group, and Birjand and Shiraz environments in the third group, which shows the similarity of the environments in the screening of genotypes in each group. In general, it is possible to see a great similarity between the genotypes of Sarigol, Hyola 401 and Okapi in the investigated environments in terms of reaction to grain yield traits. (Figure 9).

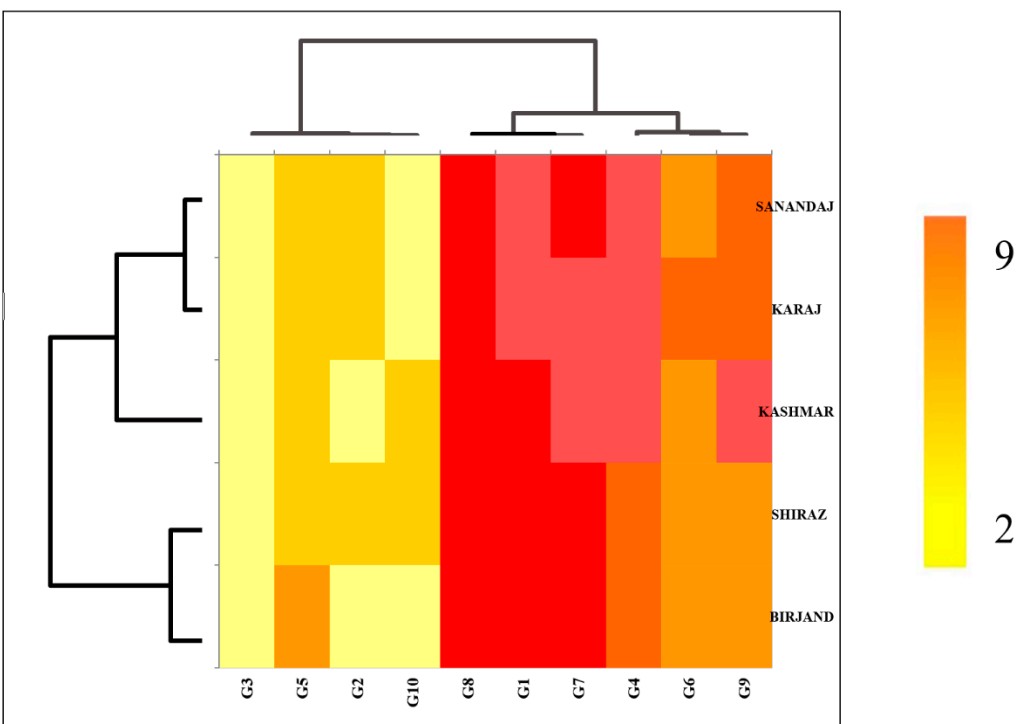

**Figure 9.** Heatmap dendrogram dividing canola genotypes into different clusters based on different environments.

## 4. Conclusions

In this research, the genotype's responses differed, i.e., genotypes are genetically distinct, which means hybridization-driven breeding programs can produce hybrid cultivars successfully. This research identified several genotypes with suitable grain yield and stability. Although these cultivars had different ratings compared to the ideal cultivar, there was also a suitable cultivar among them. Based on the results of the composite analysis, the significant variables were environment, year, year × environment, genotype × environment, year × genotype, and year × genotype × environment. Based on the graphs from the GGE biplot method, Hyola 401 and Okapi genotypes can be selected as favorable for stability under different testing conditions. Karaj was discriminatory and had high screening powers. GGE biplot graphs enabled the detection of stable cultivars in ideal environments and the grouping of cultivars and environments according to grain yield. Future breeding programs, as well as extension programs, can benefit from this research.

**Author Contributions:** S.H.S. data collection and design of the experiments; K.M., reviewing, and designed the experiments, S.H.G. designed the experiments, reviewing and editing; Á.I. completed the plant sampling in the field; C.B. wrote the manuscript; M.R.B. reviewing and editing; S.M.N.M. carried out the statistical analysis and writing the manuscript; wrote the manuscript; and J.N.; E.H. reviewed and finalized the manuscript; Visualization, S.M.N.M. and A.S.; Supervision, A.V. All authors have read and agreed to the published version of the manuscript.

**Funding:** Project no. TKP2021-NKTA-32 has been implemented with the support provided by the Ministry of Innovation and Technology of Hungary from the National Research, Development and Innovation Fund, financed under the TKP2021-NKTA funding scheme.

**Institutional Review Board Statement:** Not applicable.

**Informed Consent Statement:** Not applicable.

**Data Availability Statement:** All data supporting the conclusions of this article are included in this article.

**Conflicts of Interest:** The authors declare no conflict of interest.

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
