# Peer review of "Sustainability on Different Canola (Brassica napus L.) Cultivars by GGE Biplot Graphical Technique in Multi-Environment"

_sustainability, doi:10.3390/su15118945_

Round 1

Reviewer 1 Report

This manuscript contains standard plant breeding research with a comprehensive analysis. The manuscript needs to be improved for the following.

Comments

1.      Title: Needs improvement. There are some omissions that makes the title less meaningful.

2.      The section; lines 74-86 can be included in the discussion and compare with the results obtained in the current study.

3.      Table captions and the caption of figure 9 should be improved to be complete and to stand alone.

4.      Some language editing is required.

This manuscript contains standard plant breeding research with a comprehensive analysis. The manuscript needs to be improved for the following.

Comments

1.      Title: Needs improvement. There are some omissions that makes the title less meaningful.

2.      The section; lines 74-86 can be included in the discussion and compare with the results obtained in the current study.

3.      Table captions and the caption of figure 9 should be improved to be complete and to stand alone.

4.      Some language editing is required.

Author Response

Dear Reviewer

We appreciate your comments on improving my article.

We revised my article based on your comments. We hope it will be satisfactory for you.

Best regards.

Reviewer 2 Report

The manuscript titled “Sustainability on different Canola (Brassica napus L) Cultivars by GGE biplot Graphical Technique” by Shojaei et al tries to characterize the environmental effect on different Canola genotypes. The paper has potential but fails to meet expectations with a lack of proper explanation. I will request the authors to add the following issues- 

1.     Abstract: what is the main finding of the paper is not clear at all. Please be precise and highlight your findings.

2.     In the Introduction: Make it in a single paragraph “Yan et al. [15] showed that the performance and accuracy of the GGE biplot model 72 were higher than that of the AMMI model. 73 In one study, different rapeseed cultivars were evaluated in five regions of Chile [16]. 74 This study stated that the genotype x environment interaction significantly explained total 75 variance. The first and second principal component analyses (PC1, PC2) accounted for 76 74.5% of the total variance [16]. 77 Rahnejat et al. [17] identified Okapi, Modena, and GKH 305 genotypes as high-yield 78 and adaptable genotypes on 15 rapeseed genotypes in four different regions of Iran. Aliza- 79 deh et al. [18] investigated the genotype × environment effect to select winter rapeseed 80 varieties with a high yield on 11 rapeseed genotypes. They showed that the most favorable 81 genotypes were selected and reported regarding seed yield traits. In another study by 82 Ghasemi et al. [19], GGE biplot and AMMI methods were investigated regarding seed 83 yield traits and oil percentage in rapeseed cultivars. The results of this study showed that 84 Hyola 401, Okapi, and Sarigol cultivars had high yields in terms of seed yield and Option 85 500 and Sunday cultivars in terms of oil percentage.”.

3.     In Methods: What is the basis for choosing these 10 genotypes? However, what are the physiological parameters that were considered to check the growth pattern?

4. In results: Please discuss the results in the subheading. There is only one subheading at the end “3.1. Cluster analysis by heat map method”. Incorporate more subheadings. The figure legends are not self-explanatory and very similar to each other. Kindly differentiate them in presentation as well as writing. Recheck figure 9, where the colour scale is not matching with the heatmap colour.    

Author Response

Dear Reviewer

We appreciate your comments on improving my article.

We revised my article based on your comments. We hope it will be satisfactory for you.

  1. In Methods: What is the basis for choosing these 10 genotypes? However, what are the physiological parameters that were considered to check the growth pattern?

Answer: These genotypes were new and suitable for cultivation in the regions of Iran, and to check the stability of the cultivars, this experiment was carried out in several regions.

Best regards.

Reviewer 3 Report

The manuscript submitted by Seyed Habib Shojaei et al. identified several genotypes with suitable grain yield and stability for canola cultivars. The topics described are interesting. However, I have found some issues as follows that need to be addressed and corrected.

The authors should provide the non-abbreviated name for GGE.

How many cultivars of canola were analyzed, some say 10, some say 15 (line 87).

What do the GL and GY mean (line 141-151)? Does the GL mean location x genotype, LxG in Table 3? If so, unify the representations.

What does the df mean (Table 3)?

Figure 2 appears to be the same as Figure 1.

The six divided sections should be indicated in Figure 2. This will make it easier for the reader to understand.

In all analyses, the ranking of cultivars were G8-G7-G4-G1-G9-G5-G2-G6-G3-G10 from the top (lines 177 and 232). However, the ideal genotype analysis resulted in G8>G7>G1>G4>G9>G5>G2>G6>G3>G10 (line 298); the reason why G1 and G4 were swapped needs to be explained. Also, unify the way the rankings are written.

Minor points

Table 1 (line 92), fig 9 (line 328) should be capital letter.

To (line 181) should be small letter.

The writing style and English of the manuscript has to be checked by a professional editor.

Author Response

Dear Reviewer

We appreciate your comments on improving my article.

We revised my article based on your comments. We hope it will be satisfactory for you.

  1. the main aims of figure 9, which tell us what really?

Answer: Grouping and checking the degree of similarity of genotypes in terms of reaction to the studied trait or traits in different environments

Best regards.

Reviewer 4 Report

this paper given us a new way to understanding the interaction between genotypes and environment, could provide an significant reference for the future relevant researches. the following comments need to be considered for a better article.

1.  GCE, tell readers the full name at the first showing up.

2. the results in abstract need a deep represent, not the data list only.

3. introduction: the lines 72-89, needs integrated as their purposes.

 4. the data in table 3, showed as 69.**, please check it.

5. in all the figures, the blue and brown lines with arrows, means what ??

6. in the figure 7 and 8, the dotted circle means what?

7. the main aims of figure 9, which tell us what really?

the whole paper including the data, tables and figures, and the relation among all the materials need some carefully revised.

all the sentences need much more revised for a better understanding.

Author Response

(The authors gave the same response as above.)

Reviewer 5 Report

Dear Editor 

Thank you so much for choosing me to review this manuscript.

"Sustainability on different Canola (Brassica napus L) Cultivars by GGE biplot Graphical Technique"

Authors should show the novelty of their study. the topic is repeated.

The authors conducted this experiment to determine the stable genotypes on grain yield and suggest the most suitable genotype (s) for different environments.  the paper needs careful revision by the authors 

Please replace old references with the latest ones. At least 75% of the references of a modern manuscript should be between 2019-2023. Check your references.

all scientific names should be in italics. check the title and whole manuscript.

you need to add temperature data to the materials and methods section.

Data is good enough to justify the findings and elaborated accordingly. Proper statistical tools were used for analysis. However, the discussion section needs to be separate

The conclusion gave sufficient information about the authors' results

Author Response

(The authors gave the same response as above.)

Round 2

Reviewer 2 Report

The authors have addressed all the queries satisfactorily. 

Author Response

Dear Reviewer

Thanks for your comments to improve our article.

best regards

Reviewer 3 Report

Most of my concerns have been remedied in the revised manuscript.

The manuscript should be proofread by a professional editor.

Author Response

(The authors gave the same response as above.)

Reviewer 4 Report

1. line 77-78, one sentence as one paragraph? can it be improved? 

2. AEC view in line 156, what is the means of AEC ? the average environment. 

3. line 321, why here? 

4. Karaj> Birjand> Kashmar> Sanandaj> Shiraz in line 240-241,  G10<G3<G6<G2<G5<G9<G1<G4<G7<G8 in lines 250, G8>G7>G1>G4>G9>G5>G2>G6>G3>G10 in line 321, could you improved the expression style, using data or anova analysis, "< or >" decreased the paper level, please consider.

3. 

English language needs more recheck and improving for a better understanding.

Author Response

Dear Reviewer

Thanks for your comments to improve our article.

  1. Line 77-78, one sentence as one paragraph? can it be improved? 

Thanks for your comment. We revised

  1. AEC view in line 156, what is the means of AEC ? the average environment. 

Thanks for your comment. AEC is the average-environment coordination (AEC). I revised in the article.

  1. Line 321, why here? 

Thanks for your comment. We revised to easy understanding in article. Line 321 showed that ranking Genotype based on PCA analysis.

  1. Karaj> Birjand> Kashmar> Sanandaj> Shiraz in line 240-241,  G10<G3<G6<G2<G5<G9<G1<G4<G7<G8 in lines 250, G8>G7>G1>G4>G9>G5>G2>G6>G3>G10 in line 321, could you improved the expression style, using data or anova analysis, "< or >" decreased the paper level, please consider.

Thanks for your comment. Our analysis is based on PCA analysis. We have AEC in the figure; genotype distance from AEC showed the rank of the best genotype or environment. Each genotype or environment close to AEC showed that this genotype or environment had stability or the best performance.

Best regards

Reviewer 5 Report

After submitting the revised manuscript, all is fine 

Regards

Author Response

(The authors gave the same response as above.)
